# Mechanical Properties and Oxidation Behavior of TaWSiN Films

**DOI:** 10.3390/ma15228179

**Published:** 2022-11-17

**Authors:** Chin-Han Tzeng, Li-Chun Chang, Yung-I Chen

**Affiliations:** 1Department of Optoelectronics and Materials Technology, National Taiwan Ocean University, Keelung City 202301, Taiwan; 2Department of Materials Engineering, Ming Chi University of Technology, New Taipei 243303, Taiwan; 3Center for Plasma and Thin Film Technologies, Ming Chi University of Technology, New Taipei 243303, Taiwan; 4Center of Excellence for Ocean Engineering, National Taiwan Ocean University, Keelung City 202301, Taiwan

**Keywords:** co-sputtering, hard coatings, mechanical properties, oxidation, residual stress

## Abstract

This study explored the structural characteristics, mechanical properties, and oxidation behavior of W-enriched TaWSiN films prepared through co-sputtering. The atomic ratios [W/(W + Ta)] of the as-deposited films maintained a range of 0.77–0.81. The TaWSiN films with a Si content of 0–13 at.% were crystalline, whereas the film with 20 at.% Si was amorphous. The hardness and Young’s modulus of crystalline TaWSiN films maintained high levels of 26.5–29.9 GPa and 286–381 GPa, respectively, whereas the hardness and Young’s modulus of the amorphous Ta_7_W_33_Si_20_N_40_ films exhibited low levels of 18.2 and 229 GPa, respectively. The oxidation behavior of TaWSiN films was investigated after annealing at 600 °C in a 1%O_2_–Ar atmosphere, and cone-like Ta_0.3_W_0.7_O_2.85_ oxides formed and extruded from the TaWSiN films.

## 1. Introduction

Transition metal nitride films are widely applied for the purpose of surface modification. Reactive sputtering is a common physical vapor deposition technique for fabricating nitride films. However, both equilibrium and metastable phase structures are formed during sputtering [1]. In a Ta–N system, cubic TaN_0.1_, orthorhombic Ta_4_N, hexagonal Ta_2_N, face-centered cubic (fcc) TaN, hexagonal TaN, and body-centered tetragonal TaN*_x_* were fabricated at specific nitrogen flow ratios and temperatures [1]. In contrast, the W–N films formed an fcc W_2_N phase [2,3] and a hexagonal WN phase [4,5]. The stoichiometric ratios (*x* = N/M) of the MN*_x_* compounds were 1.0 and 0.5 for TaN and W_2_N, respectively. However, defect structures were complex for these thin-film materials, which affected their elemental compositions and stoichiometric ratios [6]. Therefore, the reported TaN*_x_* films showed deviated hardness levels of 14.7–39.5 GPa [7,8,9]. In the work of Li et al. [8], the hardness of TaN*_x_* films decreased from 39.5 to 24.7 GPa as the *x* ratio increased from 0.95 to 1.21. In the work of Yang et al. [9], the hardness of TaN*_x_* films decreased from 26.0 to 14.7 GPa as the *x* ratio increased from 1.03 to 1.53. Lou et al. [2] reported that the hardness of WN*_x_* films increased from 26.7 to 31.6 GPa when the *x* ratio increased from 0.36 to 0.43, and then decreased to 28.4 GPa when the *x* ratio increased further to 0.60. Hones et al. [4] reported that the hardness of W_2_N films decreased from 32 to 30 to 25 GPa when the *x* ratio increased from 0.3 to 0.6 to 0.7. The increase in the *x* ratio of the transition metal nitride decreased the hardness due to the effect of the increased valence electron concentration and decreased cohesive energy [4,10]. Moreover, the mechanical properties of binary nitride were affected by their phase constitution, grain size, and residual stress. Solid solution strengthening plays a vital role in enhancing the mechanical properties of ternary nitride films such as TaAlN [11], TaZrN [12], and TaWN [13]. In a previous study [14], co-sputtered TaWN films formed fcc solid solutions. The fcc structure varied from TaN-dominant to W_2_N-dominant when the [W/(Ta + W)] ratio reached a critical value of approximately 0.6. The Ta_17_W_55_N_28_ and Ta_8_W_68_N_24_ films with [W/(Ta + W)] ratios of 0.76 and 0.89, respectively, formed a W_2_N-dominant structure and exhibited a high hardness level of 29.9–31.9 GPa and a high Young’s modulus of 381–391 GPa. In contrast, transition metal nitride films with Si addition formed nanocomposite structures, such as TiSiN [15] and NbSiN [16] films, and exhibited outstanding mechanical properties. The aforementioned nanocomposite structure consisted of nanocrystalline transition metal nitride grains surrounded by amorphous SiN*_x_* tissue, which hindered the dislocation movement and raised the hardness [15,17]. Further increasing the Si content in MSiN films changed the nanocomposite structure to be amorphous due to the high amorphous SiN*_x_* volume in the films, accompanied by a decrease in hardness. However, the nanocomposite model was not applied to most transition metal nitride films [18]. Additionally, a popular observation reported in the literature is the formation of columnar structures with amorphous SiN*_x_* interlaid in the grain boundaries, such as in TaSiN [19] and WSiN [20] films. TaWN films with hardness values of 28–38 GPa have been applied as hard coatings [13,21]. Wada [22] reported that TaWSiN films exhibited a hardness of 26 GPa, which was higher than 23 GPa for the TaWN films, and suggested their potential application in cemented carbide cutting tools. Therefore, exploring the mechanical properties of TaWSiN films with structural variation in correlation with that observed in TaWN films is vital to assist in the evaluation of TaWSiN films as hard coatings.

Another objective of adding Si into nitride films is to advance oxidation resistance for high-temperature applications [20,23,24]. The amorphous structure of MSiN films and the formation of a surficial Si-oxide layer above them after annealing enhance the oxidation resistance of MSiN films. In our previous study [25], TaSi_19–21_N films exhibited oxidation resistance at 600 °C in air. With >24 at.% Si, WSiN films exhibited restricted oxidation when annealed at 600 °C in 1% O_2_–Ar [26]. Therefore, investigating the oxidation behavior of TaWSiN films is crucial.

## 2. Materials and Methods

TaWSiN films with a Ta interlayer were fabricated through co-sputtering on Si substrates. The TaWSiN films were classified into Batch S and Batch T samples. Batch S samples were prepared by applying a DC power of 100 W to the Ta target (*P*_Ta_), a DC power of 200 W to the W target (*P*_W_), and DC power levels of 0, 50, 100, and 150 W to the Si target (*P*_Si_). Batch T samples were fabricated by applying a *P*_Ta_:*P*_W_ ratio of 1:2, in which *P*_Ta_ was set at 50, 75, 85, and 100 W, with a fixed *P*_Si_ of 150 W.

The chemical compositions and elemental mappings of TaWSiN films were determined using a field-emission electron probe microanalyzer (FE-EPMA, JXA-iHP200F, JEOL, Akishima, Japan). The thicknesses of TaWSiN films and interlayers were examined using a field emission scanning electron microscope (SEM, S4800, Hitachi, Tokyo, Japan). The phases in the films were verified using X-ray diffraction (XRD; X’Pert PRO MPD, PANalytical, Almelo, The Netherlands). The determinations of lattice constants, texture coefficients, and average grain sizes of the films are described in [14]. The films’ nanostructures were examined using a transmission electron microscope (TEM, JEM-2010F, JEOL, Akishima, Japan). The TEM samples with protective C and Pt layers were prepared using a focused ion beam system (NX2000, Hitachi, Tokyo, Japan). The hardness and Young’s modulus of the films were measured using a nanoindentation tester (TI-900 Triboindenter, Hysitron, Minneapolis, MN, USA) and calculated using the Oliver–Pharr method [27]. The residual stress on the films was determined using the curvature method [28]. The average surface roughness (*Ra*) values were determined with an atomic force microscope (AFM, Dimension 3100 SPM, Nanoscope IIIa, Veeco, Santa Barbara, CA, USA).

## 3. Results and Discussion

### 3.1. Chemical Composition and Phase Structure of TaWSiN Films

Table 1 presents the sputtering power parameters and the chemical compositions of the fabricated TaWSiN films. The Batch S samples, namely, S0, S1, S2, and S3, exhibited chemical compositions of Ta_17_W_55_N_28_, Ta_14_W_56_Si_3_N_27_, Ta_13_W_49_Si_5_N_33_, and Ta_13_W_43_Si_8_N_36_ when the applied *P*_Si_ values were 0, 50, 100, and 150 W, respectively, whereas the *P*_Ta_ and *P*_W_ levels were set at 100 and 200 W, respectively. The Batch T samples were prepared to increase the Si content of the TaWSiN films further. The Batch T samples, namely, T4, T3, T2, and T1, exhibited chemical compositions of Ta_13_W_43_Si_8_N_36_, Ta_11_W_42_Si_11_N_36_, Ta_10_W_38_Si_13_N_39_, and Ta_7_W_33_Si_20_N_40_ when the applied *P*_Ta_–*P*_W_ values were 100–200, 85–170, 75–150, and 50–100 W, respectively, and the *P*_Si_ value was set at 150 W. The samples S1 and S2 were partially detached after they were prepared and were stored for weeks (as shown in Figure 1). This is attributed to their high residual stress values after co-sputtering. Figure 1 shows the secondary electron image (SEI) and elemental mappings of Ta, W, Si, N, and O. Position 2 on the SEI exhibited chemical compositions of 14.0% Ta, 57.4% W, 2.4% Si, 25.8% N, and 0.4% O, which is comparable to that of the S1 sample (listed in Table 1). The aforementioned result indicated that position 2 was located on the undetached part of the S1 film. Positions 1 and 3 on the SEI image exhibited compositions of 8.6% Ta, 38.0% W, 38.5% Si, 14.9% N, and 0.0% O and 10.5% Ta, 42.3% W, 29.9% Si, 17.3% N, and 0.0% O, respectively, belonging to the partially detached parts of the S1 films. Position 4 on the SEI image indicated the exposed Si substrate with a high density, as found by the Si mapping. Moreover, the S1 and S2 films prepared with a TaN interlayer on Si substrates exhibited residual stresses of −4.5 and −4.8 GPa, respectively, and these films did not detach for months. The data for the S1 and S2 samples in Table 1 were collected from the undetached parts of the samples prepared on Si substrates. Figure 2 depicts the variations between the chemical compositions and the Si content of the TaWSiN films. The Ta and W content decreased with increasing Si content, whereas the N content increased with increasing Si content; this implies the high affinity between N and Si. Within the Batch S samples, the Si content of TaWSiN films increased from 0 to 2.9 at.%, 4.8 at.%, and 7.7 at.% when the *P*_Si_ increased from 0 to 50, 100, and 150 W. Moreover, the Si content of the TaWSiN films increased further from 7.7 at.% to 10.3 at.%, 12.4 at.%, and 19.7 at.% for the T4, T3, T2, and T1 samples, respectively. The atomic ratios W/(W + Ta) of the S0, S1, S2, S3(T4), T3, T2, and T1 samples were 0.77, 0.80, 0.79, 0.77, 0.79, 0.80, and 0.81, respectively. These TaWSiN films exhibited a narrow W/(W + Ta) range of 0.77–0.81, which implies that the crystalline nitride phase was the same as for these films. In [14], the Ta_17_W_55_N_28_ film crystallized in an fcc phase with a W_2_N-dominant structure. *N*_cal_, the calculated N content, was defined using the stoichiometric N:M ratios of 1:2, 1:2, and 4:3 for Ta_2_N, W_2_N, and Si_3_N_4_, respectively. N/*N*_cal_, the ratio of realistic N content to *N*_cal_ was 0.77, 0.71, 0.83, 0.94, 0.90, 0.97, and 0.85 for the S0, S1, S2, S3, T3, T2, and T1 samples, respectively, implying that these TaWSiN films were N-deficient. Thus, they are comparable to the co-sputtered CrWSiN films reported in [29] due to the low affinity between N and W.

Figure 3a displays the grazing incidence XRD (GIXRD) patterns of the Batch S samples scanned at 2.4 degree/min. All the Batch S samples exhibited an fcc phase, and these reflections were located between the standard values of the TaN (ICDD 00-049-1283) and W_2_N (ICDD 00-025-1257) phases, implying the formation of nitride solid solutions. Figure 3b exhibits the GIXRD patterns of the Batch T samples. The T4, T3, and T2 films crystallized into an fcc phase, whereas the T1 films were amorphous. Figure 4a shows the cross-sectional TEM (XTEM) image of a partially detached S2(Ta_13_W_49_Si_5_N_33_) film displaying a columnar structure. The selected area electron diffraction (SAED) pattern exhibited an fcc structure with (111), (200), and (220) diffraction spots. Figure 4b displays a high-resolution TEM (HRTEM) image at the free surface of the S2 film covered with a protective C layer. The crystalline regions exhibit lattice fringes accompanied with d-spacing values of 0.215–0.221 and 0.244 nm for the (200) and (111) planes, respectively. Figure 5a displays the XTEM image of the T1(Ta_7_W_33_Si_20_N_40_) film. The SAED pattern exhibits vague rings, indicating an amorphous structure. No evident crystalline morphology was observed in either the XTEM or HRTEM (Figure 5b) images. Figure 6a depicts the lattice constants of TaWSiN films determined using the reflections in GIXRD patterns. The lattice constants increased from 0.4242 nm for the Ta_17_W_55_N_28_ film to 0.4247 and 0.4257 nm for the Ta_14_W_56_Si_3_N_27_ and Ta_13_W_49_Si_5_N_33_ films, respectively, and then decreased to 0.4252, 0.4252, and 0.4238 nm for the Ta_13_W_43_Si_8_N_36_, Ta_11_W_42_Si_11_N_36_, and Ta_10_W_38_Si_13_N_39_ films, with the Si content in the TaWSiN films increasing incessantly. This variation was attributed to the loose structure and restricted Si solubility in the TaWN matrix. Because the W_2_N phase was an fcc structure, and half of the octahedral interstitial sites in the cubic close-packed W array were empty [30], Si atoms of low addition occupied the unfilled interstitial sites, resulting in the lattice expanding. The increase in Si content resulted in amorphous SiN*_x_* forming, which gathered the Si and N atoms in the second phase or dissolved Si into the W_2_N lattice [31]. The atomic radii of Ta, W, and Si are 0.1430, 0.1367, and 0.1153 nm [32], respectively; therefore, a high Si content tends to decrease the lattice constants of TaWSiN solid solutions. Figure 6b is the modification of Figure 5 in [14] after adding the lattice constant values of the TaWSiN films. The TaWSiN films presented a W_2_N-dominant phase when the W/(W + Ta) ratio was maintained at 0.77–0.81.

### 3.2. Mechanical Properties

Table 2 lists the mechanical properties of the TaWSiN films. The Ta_17_W_55_N_28_ film exhibited a hardness (*H*) of 29.9 GPa and a Young’s modulus (*E*) of 381 GPa. The crystalline TaWSiN films exhibited *H* values of 26.5–29.5 GPa and *E* values of 286–320 GPa. The addition of Si to the TaWN films slightly decreased their mechanical properties. The amorphous Ta_7_W_33_Si_20_N_40_ film exhibited an *H* of 18.2 GPa and an *E* of 229 GPa, which were lower than those of the crystalline TaWSiN films. The aforementioned TaWSiN films exhibited mechanical properties higher than those of the TaSiN (*H*: 13.1–16.7 GPa and *E*: 158–219 GPa) [33] and WSiN films (*H*: 14.7–24.5 GPa and *E*: 226–284 GPa) [26]. The phase, texture, grain dimension, and residual stress were the vital factors that affected the mechanical properties of transition metal nitride films. In [14], the TaN-dominant TaWN films exhibited lower *H* and *E* values of 23.2–24.5 and 302–351 GPa, respectively, whereas the W_2_N-dominant TaWN films exhibited higher *H* and *E* levels of 29.9–31.9 and 381–391 GPa, respectively. In this study, all the aforementioned crystalline TaWSiN films exhibited a W_2_N-dominant structure, which sustained the *H* of TaWSiN films at a high level. Figure 7 depicts the Bragg–Brentano X-ray diffraction (BBXRD) patterns of the TaWSiN films. All the crystalline TaWSiN films exhibited a strong (200) orientation. Figure 8a illustrates the texture coefficients of the crystalline TaWSiN films. The average value of *T*c(111) and *T*c(200) was 1. The *T*c(111) decreased from 0.16 for the Ta_17_W_55_N_28_ film to 0.04 and 0.01 for the Ta_14_W_56_Si_3_N_27_ and Ta_13_W_49_Si_5_N_33_ films, respectively, and further decreased to 0 for the films with high Si levels: Ta_13_W_43_Si_8_N_36_, Ta_11_W_42_Si_11_N_36_, and Ta_10_W_38_Si_13_N_39_. Figure 8b displays the relationship between *T*c(111) and *H* of the W_2_N-dominant TaWSiN films. The Ta_8_W_68_N_24_ and Ta_17_W_55_N_28_ films studied in [14] are also shown for comparison. The films with a higher *T*c(111) exhibited higher *H* values, whereas the *H* values of the films with a *T*c(111) of 0 were diverse. The average grain sizes evaluated from the full width at half maximum reflection (200) in the BBXRD patterns were 15.6, 23.4, 24.1, 30.0, 21.0, and 23.5 nm for the Ta_17_W_55_N_28_, Ta_14_W_56_Si_3_N_27_, Ta_13_W_49_Si_5_N_33_, Ta_13_W_43_Si_8_N_36_, Ta_11_W_42_Si_11_N_36_, and Ta_10_W_38_Si_13_N_39_ films, respectively. Figure 9 displays the relationships between *H* and grain size related to the Si content. The grain size increased from 15.6 to 30.0 nm when the Si content increased from 0 to 8 at.%, which was accompanied by a decrease in *H* from 29.9 to 26.5 GPa. This variation coincided with the Hall–Petch relationship for polycrystalline materials. In contrast, the films with 11 and 13 at.% Si content exhibited lower grain sizes and higher H than the films with 8 at.% Si, which were accompanied by a higher volume of amorphous SiN*_x_*. Figure 10 depicts the hardness and average grain size of the TaWSiN films, and the data from the TaWN films [14] are also listed for comparison. The results indicate that the variations in hardness related to average grain size for crystalline TaW(Si)N films followed the Hall–Petch relationship and the aforementioned films exhibited W_2_N-dominant structures. In contrast, the variations in hardness and average grain size for crystalline TaWN films with TaN-dominant structures exhibited an inverse Hall–Petch relationship. The residual stress of the crystalline TaWSiN films was maintained at a high compressive level ranging from −3.3 to −4.2 GPa. In the work of Chang et al. [34], TiN and CrN interlayers were used to replace the Ti and Cr interlayers between the AlTiCrN coatings and SKH9 tool steels, respectively, which lowered the residual stress. The S1 and S2 films prepared with a TaN interlayer on Si substrates exhibited residual stresses of −4.5 and −4.8 GPa, *H* values of 27.7 and 26.7 GPa, and *E* values of 309 and 311 GPa, respectively. Moreover, the S3 film prepared with a TaN interlayer exhibited a residual stress of −4.1 GPa, which was comparable with −4.2 GPa for the S3 film prepared with a Ta interlayer. The aforementioned results implied that the partial detachment of S1 and S2 films with a Ta interlayer was attributable to the effect of high residual stress. The *H/E* indicator was used to evaluate the toughness of films [35]. The crystalline TaWSiN films exhibited *H/E* values of 0.084–0.100, which were higher than those of the TaWN films (0.070–0.082) [14] and the amorphous Ta_7_W_33_Si_20_N_40_ films (0.079). The *Ra* values of the crystalline TaWSiN films were in the range of 2.2–3.6 nm, which was slightly higher than those of the TaWN films (1.6–2.7 nm) [14] and the amorphous Ta_7_W_33_Si_20_N_40_ films (0.8 nm).

### 3.3. Oxidation Behavior

The crystalline T2(Ta_10_W_38_Si_13_N_39_) and amorphous T1(Ta_7_W_33_Si_20_N_40_) samples were annealed at 600 °C in 1% O_2_–Ar for 8 h. Annealing at 600 °C in 1% O_2_–Ar was utilized to evaluate the performance of the protective coatings on glass molding dies [26,33]. Extruded oxides were observed on the surfaces of the annealed TaWSiN films. Figure 11 and Figure 12 display the surficial SEM morphologies of the 8-h-annealed T2 and T1 samples, respectively. The cross-sectional SEM (XSEM) images of the T2 and T1 samples indicated that these oxides were cone-like (Figure 13). The cone extruded from the surface of the annealed T2 film and exhibited a diameter of 612 nm and a height of 495 nm, which formed a volume of approximately 0.0485 μm^3^. Moreover, the cone on the annealed T1 film was 2629 nm in diameter and 2554 nm in height, which formed a volume of 4.621 μm^3^, 95 times that of the cone on the aforementioned T2 film. Figure 14a exhibits an XTEM image of the 8-h-annealed T2 sample, and the dimensions of the cone-like oxides are twice the size of those shown in the XSEM image. The cone showed a convex surface toward the film. The SAED patterns at regions A and B of the annealed T2 sample indicated that the film was crystalline in the fcc phase with a columnar structure, whereas the cones were nanocrystalline and exhibited a Ta_0.3_W_0.7_O_2.85_ phase. Figure 14b displays a magnified XTEM image of one cone-like oxide consisting of laminated layers; each layer comprises granular grains. The layer period was approximately 20 nm. Figure 15a displays an XTEM image and an SAED pattern at region C of the annealed T1 sample. The SAED pattern at region C indicates an amorphous phase of the unoxidized film. Figure 15b shows an XTEM image at region D, which reveals a laminated structure with a stacking period of approximately 16 nm, and the SAED pattern exhibits a Ta_0.3_W_0.7_O_2.85_ phase. The common characteristics of these cone-like oxides for both the T2 and T1 films were: (1) theses oxides exhibited laminated structures and each layer comprised granular grains; (2) the grain sizes and layer periods of the interior regions were lower than those of the outer regions. The aforementioned observation showed differences from the morphology of the oxidized W_42_Si_16_N_42_ film annealed at 600 °C in 1% O_2_–Ar for 4 h, which comprised pyramid oxides with heights of hundreds of nm accompanied with lateral cracks due to the formation of WO_3_; however, no structure of laminated layers was observed [26]. Moreover, the oxidation resistance of WSiN films was improved by increasing the Si content to 24 at% due to the formation of a SiO_x_ scale. In contrast, the increase in Si content of the TaWSiN films revealed an inferior oxidation resistance. The oxidation behavior of TaWSiN films can be concluded as shown in Figure 16. At the beginning of oxidation, Ta_0.3_W_0.7_O_2.85_ oxides nucleated on the free surface of these W-enriched TaWSiN films. The Ta_0.3_W_0.7_O_2.85_ oxides grew to a dimension larger than 10 nm, and new nuclei formed at the interface between oxides and unoxidized films. New granular oxides formed and the area of the newborn interface area became larger than its preceding state. Moreover, the interface formed a curved surface. Figure 17 displays the GIXRD patterns of the annealed T2 and T1 samples, indicating that the oxides form a Ta_0.3_W_0.7_O_2.85_ [ICDD 00-045-0116] phase.

## 4. Conclusions

The crystalline TaWSiN films with a W/(W + Ta) ratio of 0.77–0.81 and a Si content of 0–13 at.% exhibited a W_2_N-dominant fcc structure with (200) texture, high hardness values of 26.5–29.9 GPa, and Young’s modulus values of 286–381 GPa. Moreover, the mechanical properties of these W_2_N-dominant TaWSiN films followed the Hall–Petch relationship according to the variation in average grain size. In contrast, the amorphous Ta_7_W_33_Si_20_N_40_ films with a high Si content of 20 at.% exhibited low hardness and Young’s modulus values of 18.2 and 229 GPa, respectively.

Cone-like Ta_0.3_W_0.7_O_2.85_ oxides formed on both the surfaces of the crystalline Ta_10_W_38_Si_13_N_39_ films and amorphous Ta_7_W_33_Si_20_N_40_ films after they were annealed at 600 °C in a 1% O_2_–Ar atmosphere. The cone oxides consisted of laminated sublayers and each sublayer comprised granular grains with diameters of 10–20 nm. Beneath these oxides, the films maintained structures similar to those of the as-deposited state.

W_2_N-dominant TaWSiN films possess superior mechanical properties to TaSiN and WSiN films, and can be utilized as hard coatings.

## Figures and Tables

**Figure 1 materials-15-08179-f001:**
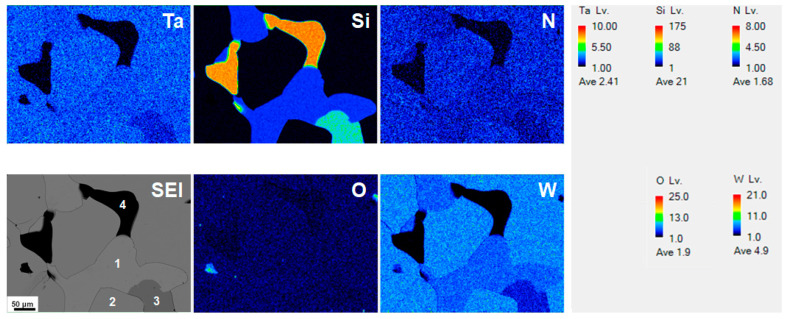
SEI and elemental mappings of the S1 sample with a Ta interlayer prepared on Si substrate.

**Figure 2 materials-15-08179-f002:**
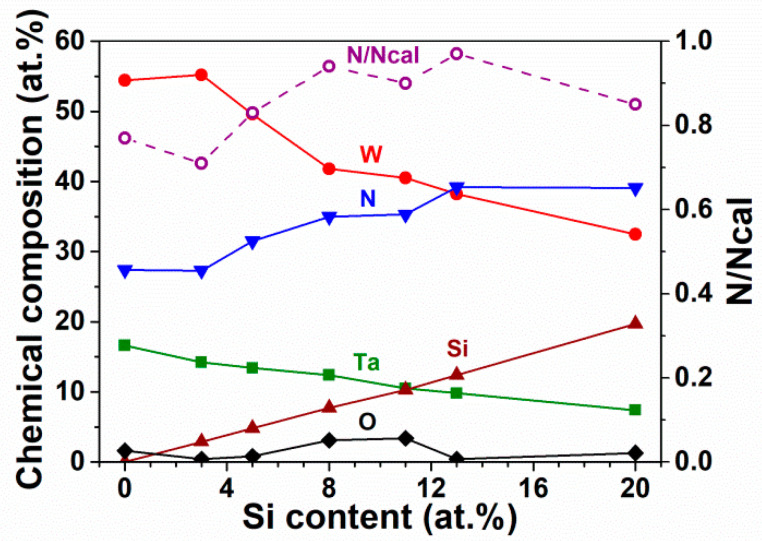
Chemical compositions and N/*N_cal_* ratios of TaWSiN films with various Si contents.

**Figure 3 materials-15-08179-f003:**
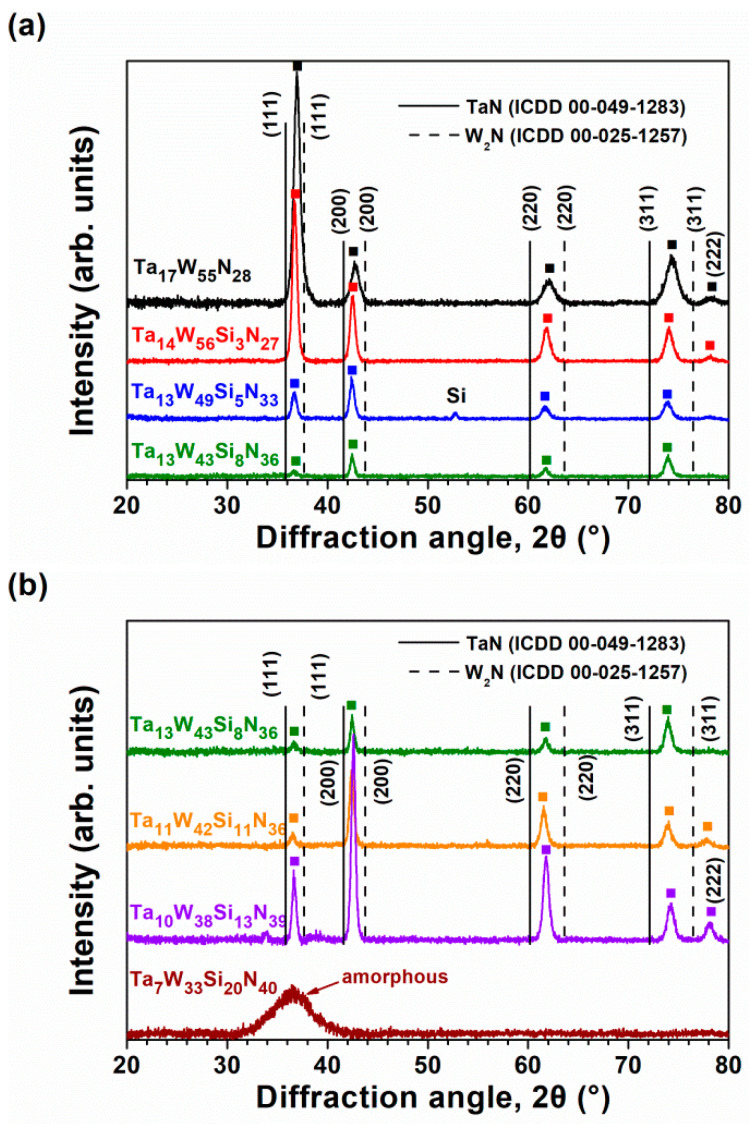
GIXRD patterns of (**a**) Batch S and (**b**) Batch T TaWSiN films.

**Figure 4 materials-15-08179-f004:**
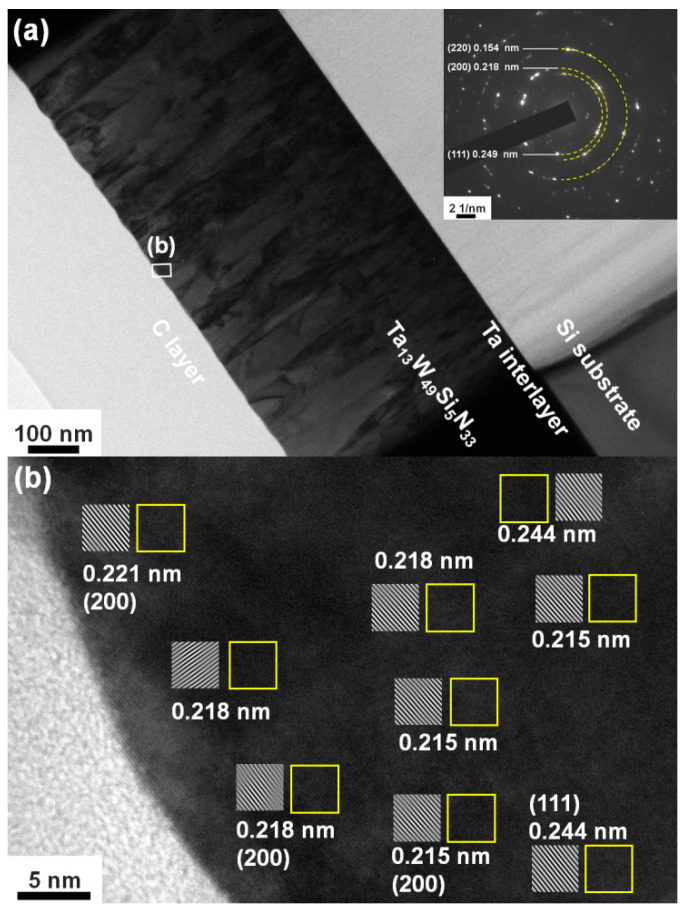
(**a**) XTEM image and SAED pattern and (**b**) HRTEM image of the partially detached Ta_13_W_49_Si_5_N_33_ film.

**Figure 5 materials-15-08179-f005:**
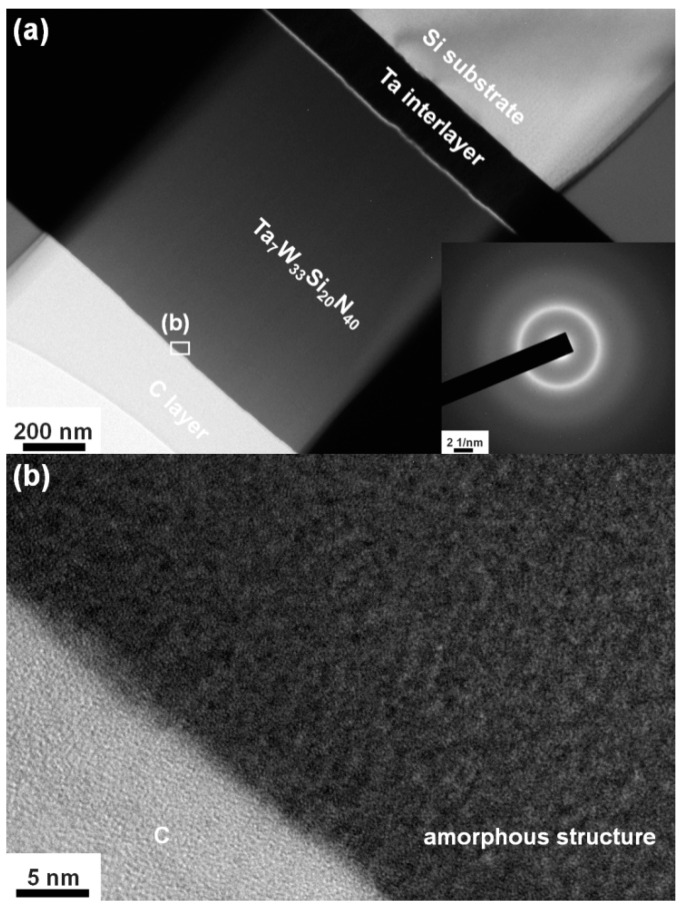
(**a**) XTEM image and SAED pattern and (**b**) HRTEM image of the Ta_7_W_33_Si_20_N_40_ film.

**Figure 6 materials-15-08179-f006:**
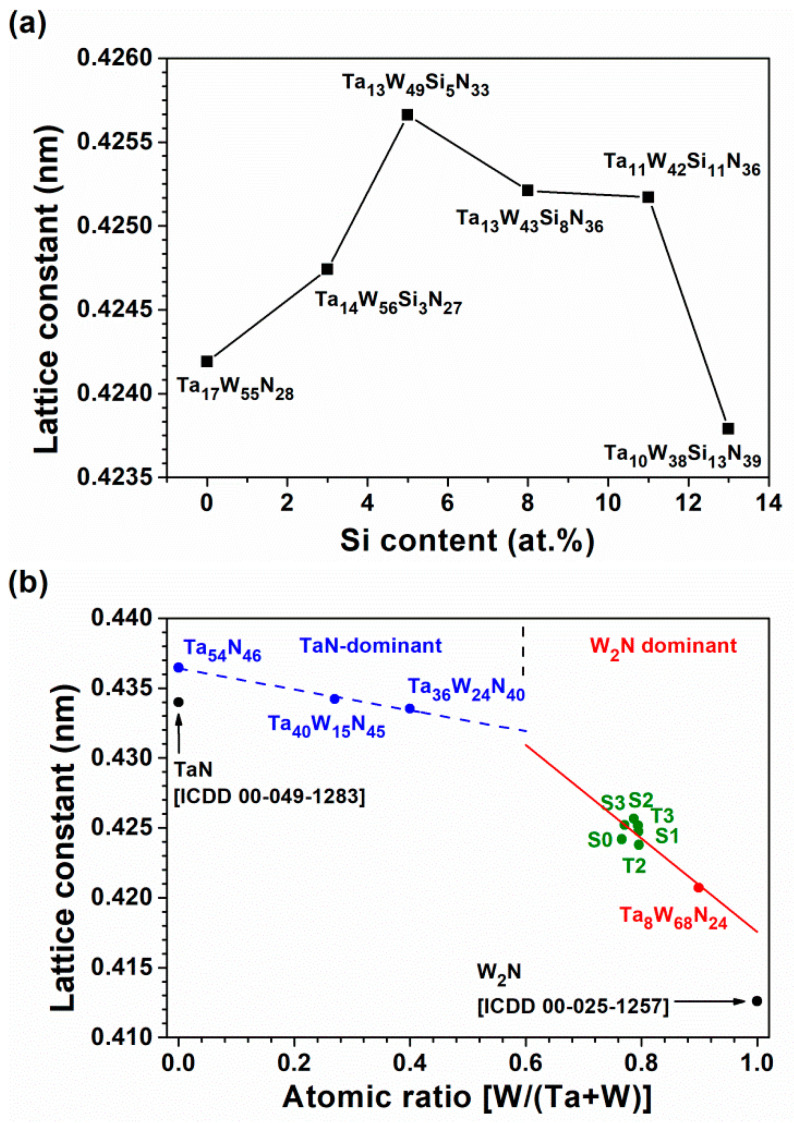
(**a**) Lattice constants of the TaWSiN films and (**b**) the relationship between lattice constants and the atomic ratio of W/(Ta + W) for TaWSiN and TaWN [14] films.

**Figure 7 materials-15-08179-f007:**
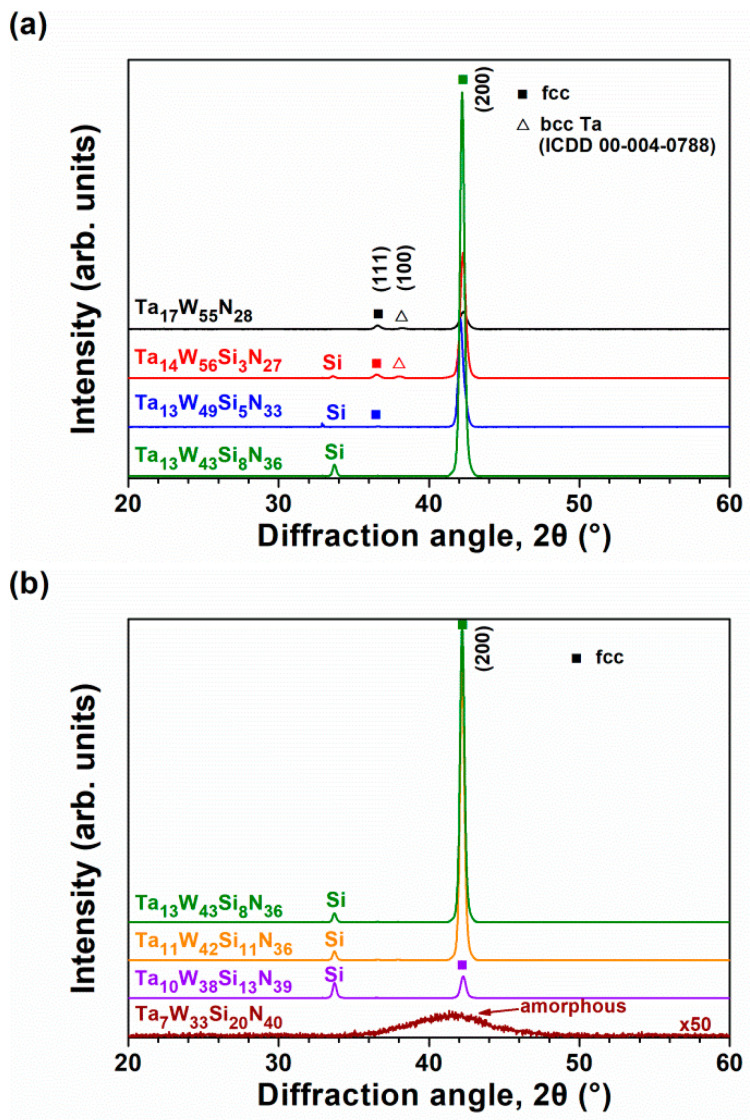
BBXRD patterns of (**a**) Batch S and (**b**) Batch T TaWSiN films.

**Figure 8 materials-15-08179-f008:**
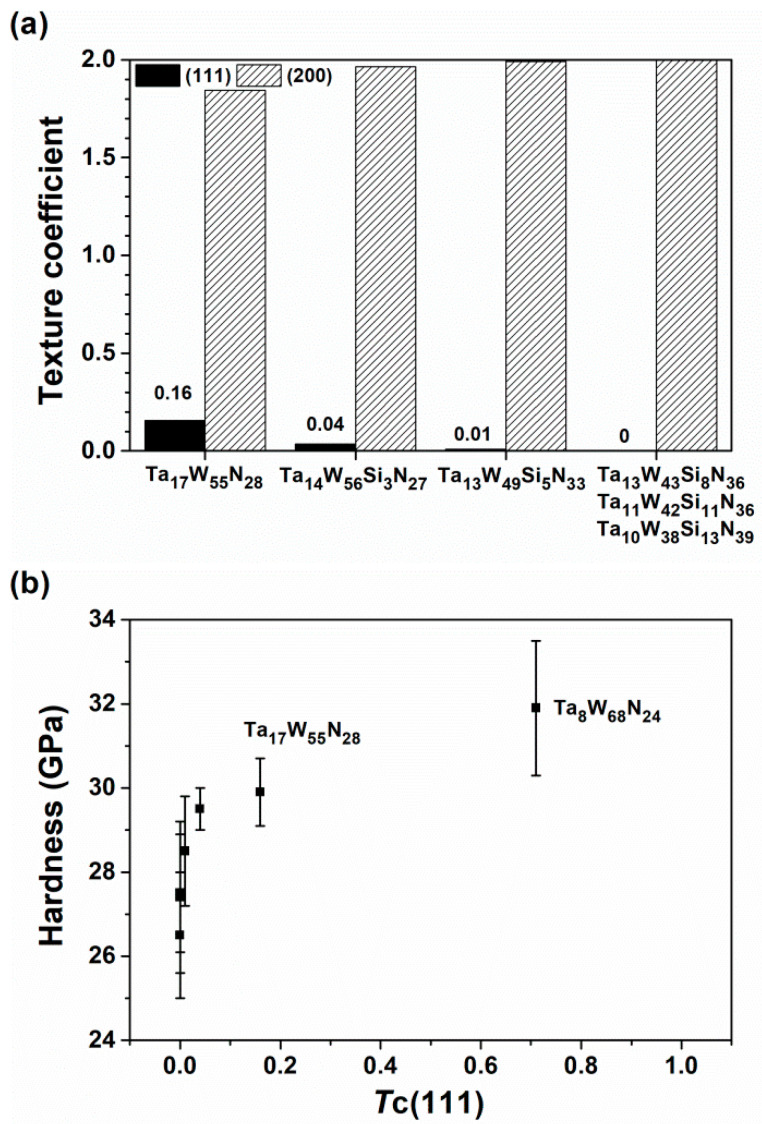
(**a**) Texture coefficient and (**b**) its relationship to the hardness of the TaWSiN films.

**Figure 9 materials-15-08179-f009:**
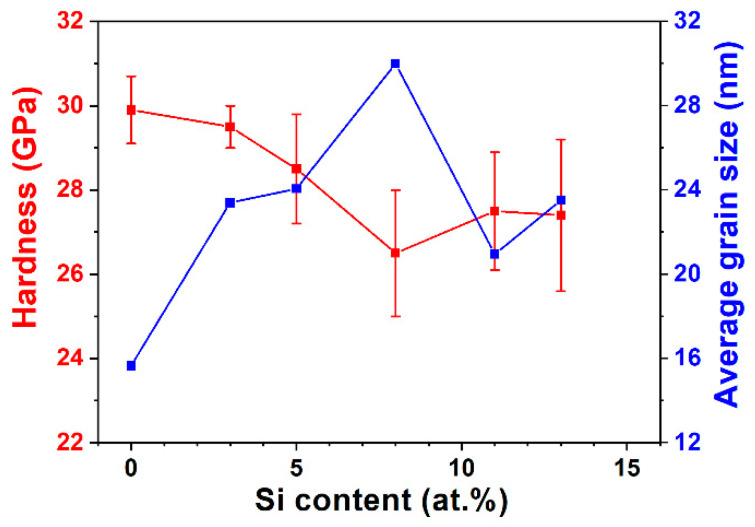
Variations in hardness and average grain size in relation to the Si content of the TaWSiN films.

**Figure 10 materials-15-08179-f010:**
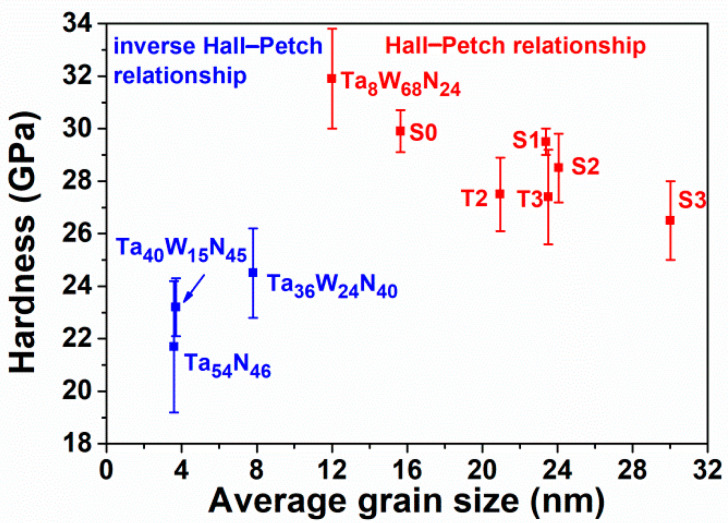
Relationship between hardness and average grain size of TaWN [14] and TaWSiN films.

**Figure 11 materials-15-08179-f011:**
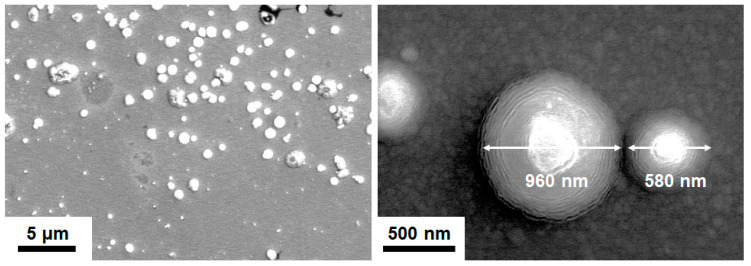
SEM images of the T2 samples annealed at 600 °C in 1% O_2_–Ar for 8 h.

**Figure 12 materials-15-08179-f012:**
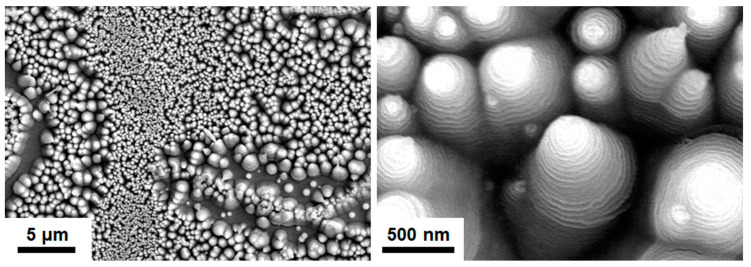
SEM images of the T1 samples annealed at 600 °C in 1% O_2_–Ar for 8 h.

**Figure 13 materials-15-08179-f013:**
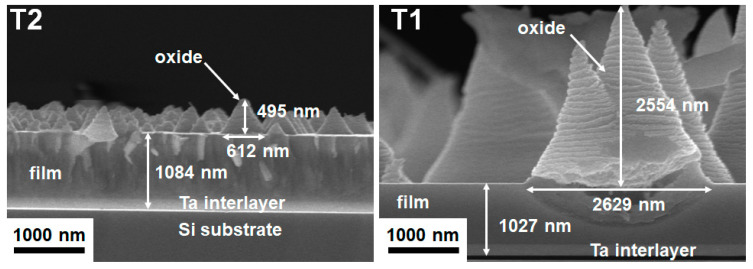
XSEM images of the T2 and T1 samples annealed at 600 °C in 1% O_2_–Ar for 8 h.

**Figure 14 materials-15-08179-f014:**
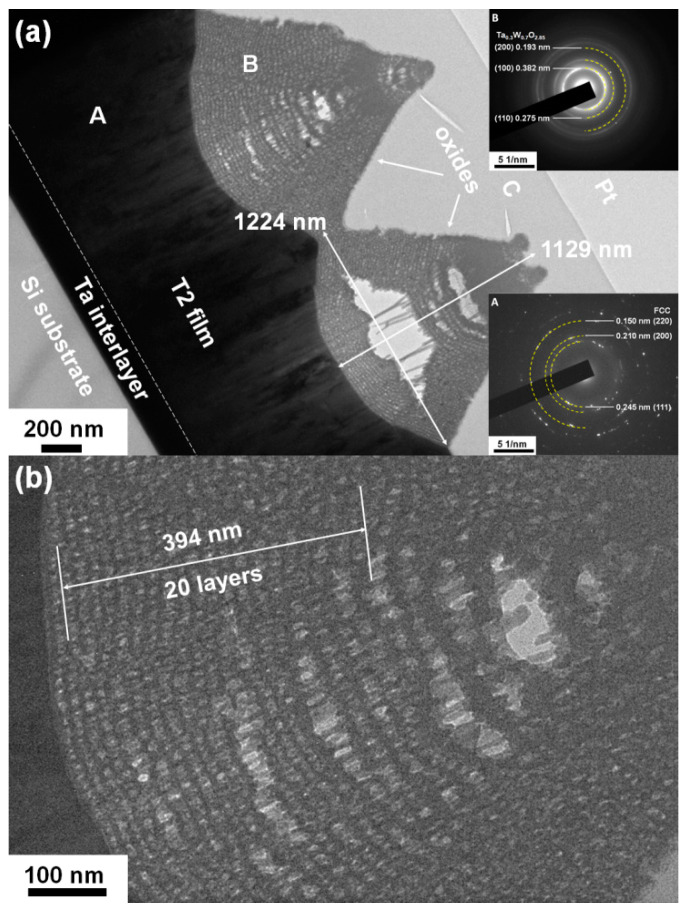
(**a**) XTEM image and SAED patterns and (**b**) magnified XTEM image of the T2 sample annealed at 600 °C in 1% O_2_–Ar for 8 h.

**Figure 15 materials-15-08179-f015:**
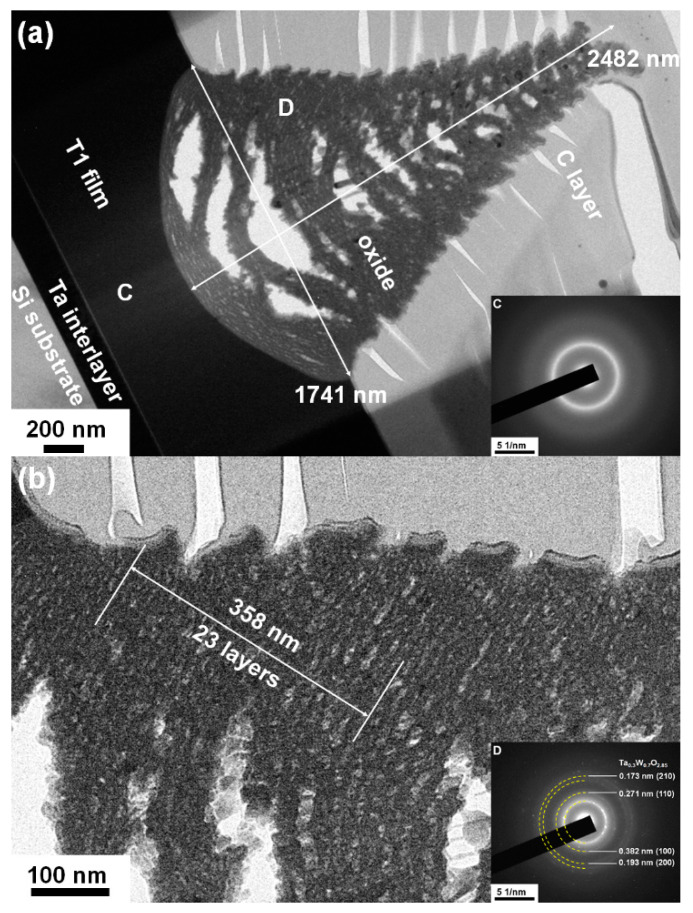
(**a**) and (**b**) XTEM images and SAED patterns of the T1 sample annealed at 600 °C in 1% O_2_–Ar for 8 h.

**Figure 16 materials-15-08179-f016:**
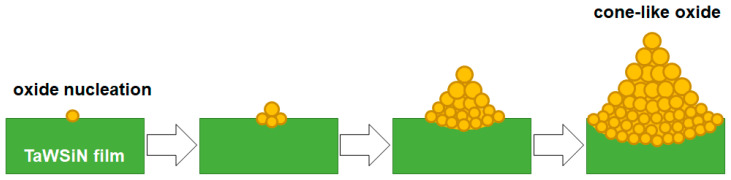
Schematic of the oxidation behavior of TaWSiN films annealed at 600 °C in 1% O_2_–Ar.

**Figure 17 materials-15-08179-f017:**
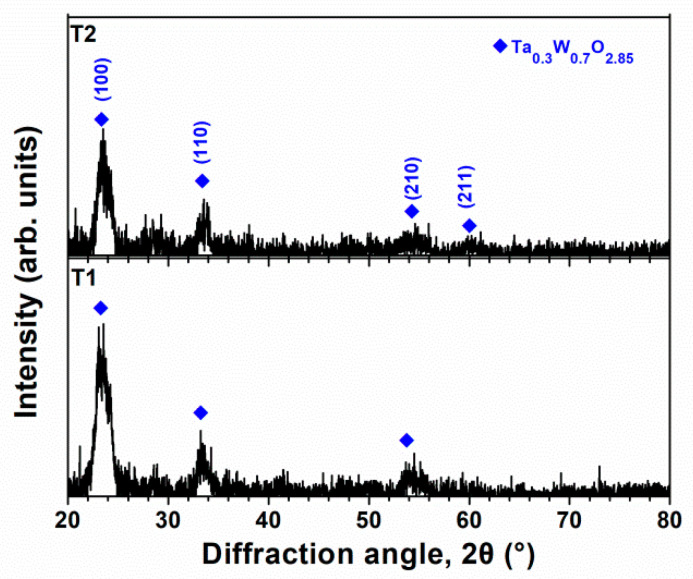
GIXRD patterns of the T2 and T1 samples annealed at 600 °C in 1% O_2_–Ar for 8 h.

**Table 1 materials-15-08179-t001:** Sputtering powers, chemical compositions, and thicknesses of the TaWSiN films.

Sample		Power (W)			Chemical Composition (at.%)					*T_F_*^1^ (nm)	*T_I_*^2^ (nm)	*R*^3^ (nm/min)
		*P* _Ta_	*P* _W_	*P* _Si_	Ta	W	Si	N	O			
S0	Ta_17_W_55_N_28_	100	200	0	16.6 ± 0.0	54.4 ± 0.0	-	27.4 ± 0.0	1.6 ± 0.0	939	157	14.4
S1	Ta_14_W_56_Si_3_N_27_	100	200	50	14.2 ± 0.3	55.2 ± 0.4	2.9 ± 0.1	27.3 ± 0.5	0.4 ± 0.4	849	110	13.1
S2	Ta_13_W_49_Si_5_N_33_	100	200	100	13.4 ± 0.7	49.6 ± 1.3	4.8 ± 0.2	31.5 ± 1.7	0.8 ± 0.2	961	103	14.8
S3(T4)	Ta_13_W_43_Si_8_N_36_	100	200	150	12.4 ± 0.3	41.8 ± 0.2	7.7 ± 0.1	35.0 ± 0.5	3.1 ± 0.0	1115	168	17.2
T3	Ta_11_W_42_Si_11_N_36_	85	170	150	10.5 ± 0.6	40.5 ± 0.9	10.3 ± 0.5	35.3 ± 0.8	3.4 ± 0.8	1178	175	18.1
T2	Ta_10_W_38_Si_13_N_39_	75	150	150	9.8 ± 0.2	38.2 ± 0.7	12.4 ± 0.3	39.2 ± 0.2	0.4 ± 0.8	870	126	13.4
T1	Ta_7_W_33_Si_20_N_40_	50	100	150	7.4 ± 0.1	32.5 ± 0.6	19.7 ± 0.5	39.1 ± 0.9	1.3 ± 1.1	767	126	11.8

^1^ *T_F_*: film thickness; ^2^ *T_I_*: Ta interlayer thickness; ^3^ *R*: deposition rate of TaWSiN film.

**Table 2 materials-15-08179-t002:** Mechanical properties of the TaWSiN/Ta/Si samples.

Sample	*H* ^1^	*E* ^2^	*H/E*	*W_e_* ^3^	*σ* ^4^	*Ra* ^5^
	(GPa)	(GPa)		(%)	(GPa)	(nm)
Ta_17_W_55_N_28_	29.9 ± 0.8	381 ± 12	0.078	63	−2.5 ± 0.1	1.6
Ta_14_W_56_Si_3_N_27_	29.5 ± 0.5	320 ± 9	0.092	63	-	2.2
Ta_13_W_49_Si_5_N_33_	28.5 ± 1.3	286 ± 8	0.100	65	-	3.4
Ta_13_W_43_Si_8_N_36_	26.5 ± 1.5	314 ± 10	0.084	62	−4.2 ± 0.0	2.9
Ta_11_W_42_Si_11_N_36_	27.5 ± 1.4	314 ± 10	0.088	66	−3.4 ± 0.1	2.7
Ta_10_W_38_Si_13_N_39_	27.4 ± 1.8	297 ± 13	0.092	60	−3.3 ± 0.0	3.6
Ta_7_W_33_Si_20_N_40_	18.2 ± 0.3	229 ± 4	0.079	54	−1.2 ± 0.1	0.8

^1^ *H*: hardness; ^2^ *E*: Young’s modulus; ^3^*W_e_*: elastic recovery; ^4^ *σ*: residual stress; ^5^ *Ra*: average surface roughness.

## Data Availability

Not applicable.

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
