# Peer review of "Mechanical Properties and Oxidation Behavior of TaWSiN Films"

_materials, 2022, doi:10.3390/ma15228179_

Round 1

Reviewer 1 Report

The paper is interesting. Please answer my comments before any decision of acceptance

1- Please provide some kind of comparison with other materials that exist in the literature concerning the mechanical properties.

2- Do you believe that annealing at 600 C for 8 hours is not practical and takes a lot of energy and time ?  ""The crystalline T2(Ta10W38Si13N39) and amorphous T1(Ta7W33Si20N40) samples were 240 annealed at 600 °C in 1% O2–Ar for 8 h""

3-Is it possible to claim that depositing a layer of siO2 or Al2O3 will not solve this problem and replace the protocol that you have developed? What is the additional value of your work? What is the benefit compared to the existing ones.

Author Response

1- Please provide some kind of comparison with other materials that exist in the literature concerning the mechanical properties.

A: The Mechanical properties of TaN and W2N films were already provided in the Introduction. New sentences “TaWN films with hardness values of 28–38 GPa have been applied as hard coatings [13,21]. Wada [22] reported that TaWSiN films exhibited a hardness of 26 GPa, which was higher than 23 GPa for the TaWN films, and suggested their potential application in cemented carbide cutting tools.” were added in lines 62–66.

A new sentence “The aforementioned TaWSiN films exhibited mechanical properties higher than those of the TaSiN (H: 13.1–16.7 GPa and E: 158–219 GPa) [33] and WSiN films (H: 14.7–24.5 GPa and E: 226–284 GPa) [26].” was added in lines 197–199.

2- Do you believe that annealing at 600 C for 8 hours is not practical and takes a lot of energy and time ?  ""The crystalline T2(Ta10W38Si13N39) and amorphous T1(Ta7W33Si20N40) samples were annealed at 600 °C in 1% O2–Ar for 8 h""

A: Thanks for the comment!  “Annealing at 600 °C in 1% O2–Ar was utilized to evaluate the performance of the protective coatings on glass molding dies [26,33].” was added in lines 261–262.

New sentences “The aforementioned observation showed differences from the morphology of the oxidized W42Si16N42 film annealed at 600 °C in 1% O2–Ar for 4 h, which comprised pyramid oxides with heights of hundreds of nm accompanied with lateral cracks due to the formation of WO3; however, no structure of laminated layers was observed [26]. Moreover, the oxidation resistance of WSiN films was improved by increasing the Si content to 24 at% due to the formation of a SiOx scale. In contrast, the increase in Si content of the TaWSiN films revealed an inferior oxidation resistance.” were added in lines 285–291.

3-Is it possible to claim that depositing a layer of SiO2 or Al2O3 will not solve this problem and replace the protocol that you have developed? What is the additional value of your work? What is the benefit compared to the existing ones.

A: Thanks for the suggestion! An additional oxide layer deposited on the TaWSiN films should not be the solution for improving the oxidation resistance of nitride films. The adhesion strength between oxide and nitride layers need to be resolved. When TaWSiN films are utilized as hard coatings at elevated temperatures, any damage on the surface could destroy the alien oxide layer. The oxide consists of elements from the below nitride films should be the way to lengthen the lifetime.

The literature reported the performance of TaWSiN was scant. Only Ref.22 had reported the mechanical properties of TaWSiN films. However, the relationship between the structural and mechanical properties was not discussed. We have clearly shown their connection in this study. Moreover, the specific oxide formation of TaWSiN films was observed.

Reviewer 2 Report

Tzeng et al. have presented the manuscript titled: Mechanical Properties and Oxidation Behavior of TaWSiN Films. In the manuscript, authors have investigated the structural characteristics, mechanical properties, and oxidation behavior of W-enriched TaWSiN films. Overall presentation of the proposed work is good, but there requiring some modifications which I think are necessary to explain before publication.

1.       There must be last single sentence about the application of the fabricated material where it has the potential utilization to make the attraction for the readers.

2.       About XRD analysis, how authors have calculated the lattice parameters described? Have they used some software like fullprof or material studio?

3.       Authors have described that there occurred the strong change in the lattice constants from Ta17W55N28 (0.4242) to Ta13W49Si5N33 (0.4252) with increasing trend, and then decreased to (0.4238 nm) for Ta10W38Si13N39. Authors have attributed this change to the “loose structure and 161 restricted Si solubility in TaWN matrix”, for which sense decreasing or increasing trend? I suggest focus on the atomic radii of elements, moreover the change in Ta, W and N contents are other reasons to cause this.

4.       This structural distortion can best be described if authors can simulate their XRD patterns and then analyze it.

5.       Moreover the intensities of the peaks are also strange, authors have described that for films with Si contents of 0–13 at.% were crystalline, then at what scan rate they have checked the XRD. I suggest to at least scanning the samples for 10 minutes.

6.       I admire the study authors have performed for this work, but these are just the characterizations of the fabricated films. Authors have not focused on the properties with application point of view. Do these samples have fewer applications?

7.       Please add the last sentence in the conclusion regarding the practical application where material can be utilized.

Author Response

1. There must be last single sentence about the application of the fabricated material where it has the potential utilization to make the attraction for the readers.

A: Hard coatings is the main application of TaWSiN films.

New sentences “TaWN films with hardness values of 28–38 GPa have been applied as hard coatings [13,21]. Wada [22] reported that TaWSiN films exhibited a hardness of 26 GPa, which was higher than 23 GPa for the TaWN films, and suggested their potential application in cemented carbide cutting tools. Therefore, exploring the mechanical properties of TaWSiN films with structural variation in correlation with that observed in TaWN films is vital to assist in the evaluation of TaWSiN films as hard coatings” were added in lines 62–68.

2. About XRD analysis, how authors have calculated the lattice parameters described? Have they used some software like fullprof or material studio?

A: “The determinations of lattice constants, texture coefficients, and average grain sizes of the films are described in [14].” was mentioned in lines 88 and 89.

In [14], below messages were described.

The lattice constants, ɑ0, of each film were evaluated according to the following equation:

, (1)

where É‘ is the lattice constant for the distinct reflection, K is the constant, and θ is the diffraction angle. The Bragg–Brentano scan (θ–2θ scan) mode was employed to determine the texture coefficients [20] and grain size of the films [34]. The texture coefficient (Tc) was defined as

, (2)

where Im(hkl) is the measured relative intensity of the reflection from the (hkl) plane, I0(hkl) is the relative intensity from the same plane in a standard reference sample, and n is the total number of reflection peaks from the coating. The grain size was calculated as

, (3)

where D is the grain size, λ is the X-ray wavelength, β is the full width at half maximum (FWHM) of reflection, and θB is the Bragg angle.

3. Authors have described that there occurred the strong change in the lattice constants from Ta17W55N28 (0.4242) to Ta13W49Si5N33 (0.4252) with increasing trend, and then decreased to (0.4238 nm) for Ta10W38Si13N39. Authors have attributed this change to the “loose structure and restricted Si solubility in TaWN matrix”, for which sense decreasing or increasing trend? I suggest focus on the atomic radii of elements, moreover the change in Ta, W and N contents are other reasons to cause this.

A: Thanks for the suggestion! Related illustrations were modified as “…which gathered the Si and N atoms in the second phase or dissolved Si into the W2N lattice [31]. The atomic radii of Ta, W, and Si are 0.1430, 0.1367, and 0.1153 nm [32], respectively; therefore, a high Si content tends to decrease the lattice constants of TaWSiN solid solutions.” in lines 173–177.

4. This structural distortion can best be described if authors can simulate their XRD patterns and then analyze it.

A: The lattice constants of TaWSiN solid solution were calculated from XRD patterns as described in the response to Comment 2.

5. Moreover the intensities of the peaks are also strange, authors have described that for films with Si contents of 0–13 at.% were crystalline, then at what scan rate they have checked the XRD. I suggest to at least scanning the samples for 10 minutes.

A: The scan rate was 2.4 degree per min. It took 25 min to finish the XRD test for phase identification. Related modification “ …, scanned at 2.4 degree/min.” was added in line 150.

6. I admire the study authors have performed for this work, but these are just the characterizations of the fabricated films. Authors have not focused on the properties with application point of view. Do these samples have fewer applications?

A: Thanks for the suggestion! Please read the response to comment 7. The goal for applying as hard coatings has been achieved. Further advancing the oxidation resistance of the TaWSiN films should be the critical issue.

7. Please add the last sentence in the conclusion regarding the practical application where material can be utilized.

A: The new sentence “W2N-dominant TaWSiN films possess superior mechanical properties to TaSiN and WSiN films, and can be utilized as hard coatings.” was added in lines 331–332.  

Reviewer 3 Report

[1]         Keywords: write 5

[2]         Introduction: The objective of the present work carefully.

[3]         Results and Discussion:

Why the author didn’t measure the thermal conductivity, impact and roughness values of the samples?

Why the author didn’t measure the other mechanical properties such as compression strength of the samples?

Why the author didn’t measure the other optical properties of the samples?

[4]   What does it add to the subject area compared with other published material?

[5]   Do you consider the topic original or relevant in the field? Does it address a specific gap in the field?

[6]   What is the main question addressed by the research?

[7]         References: cite the following recent references

DOI: https://doi.org/10.1007/s11082-016-0812-7

DOI: https://doi.org/10.1007/s11082-017-1030-7

Author Response

[1]         Keywords: write 5

A: “hard coatings” and “residual stress” were added.

[2]         Introduction: The objective of the present work carefully.

A: The revised part “TaWN films with hardness values of 28–38 GPa have been applied as hard coatings [13,21]. Wada [22] reported that TaWSiN films exhibited a hardness of 26 GPa, which was higher than 23 GPa for the TaWN films, and suggested their potential application in cemented carbide cutting tools. Therefore, exploring the mechanical properties of TaWSiN films with structural variation in correlation with that observed in TaWN films is vital to assist in the evaluation of TaWSiN films as hard coatings.” was shown in lines 62-68.

[3]         Results and Discussion:

Why the author didn’t measure the thermal conductivity, impact and roughness values of the samples?

A: This study aimed on evaluating the mechanical properties and oxidation resistance of TaWSiN films. Thermal conductivity and thermal impact were not the topic. Average surface roughness values were added in Table 2 and illustrated as “The Ra values of the crystalline TaWSiN films were in the range of 2.2–3.6 nm, which was slightly higher than those of the TaWN films (1.6–2.7 nm) [14] and the amorphous Ta7W33Si20N40 films (0.8 nm).” in lines 244–246.

“The average surface roughness (Ra) values were determined with an atomic force microscope (AFM, Dimension 3100 SPM, Nanoscope IIIa, Veeco, Santa Barbara, CA, USA).” was added in lines 96–98.

Why the author didn’t measure the other mechanical properties such as compression strength of the samples?

A: “The H/E indicator was used to evaluate the toughness of films [35]. The crystalline TaWSiN films exhibited H/E values of 0.084-0.100, which were higher than those of the TaWN films (0.070–0.082) [14] and the amorphous Ta7W33Si20N40 films (0.079).” was added in lines 241–243.

Why the author didn’t measure the other optical properties of the samples?

A: TaWSiN films were not transparent. Optical properties were not the objective in this study.

[4]   What does it add to the subject area compared with other published material?

A: The new sentence “W2N-dominant TaWSiN films possess superior mechanical properties to TaSiN and WSiN films, and can be utilized as hard coatings.“ was added in lines 331–332.

[5]   Do you consider the topic original or relevant in the field? Does it address a specific gap in the field?

A: The literature reported the performance of TaWSiN was scant. Only Ref.22 (response to comment 2) had reported the mechanical properties of TaWSiN films. However, the relationship between the structural and mechanical properties was not discussed. We have clearly shown their connection in this study. Moreover, the specific oxide formation of TaWSiN films was observed.

[6]   What is the main question addressed by the research?

A: The effects of substrate bias and temperature on the mechanical properties of TaWSiN films should be investigated in a future work.

[7]         References: cite the following recent references 

DOI: https://doi.org/10.1007/s11082-016-0812-7

DOI: https://doi.org/10.1007/s11082-017-1030-7

A: Thanks for the suggestion! However, these two papers investigated the electrical conduction mechanisms for Au/n+(a-Si:H)/n(SiGe)/p(c-Si)/Ag heterojunction and optical properties for Al:ZnO films, respectively, which were not related to this study and should not be cited.